# Mediators of Amylin Action in Metabolic Control

**DOI:** 10.3390/jcm11082207

**Published:** 2022-04-15

**Authors:** Christina N. Boyle, Yi Zheng, Thomas A. Lutz

**Affiliations:** Institute of Veterinary Physiology, Vetsuisse Faculty University of Zurich, Winterthurerstrasse 260, 8057 Zurich, Switzerland; boyle@vetphys.uzh.ch (C.N.B.); yi.zheng@uzh.ch (Y.Z.)

**Keywords:** amylin, satiating, glucagon, calcitonin receptor, receptor-activity modifying protein, leptin

## Abstract

Amylin (also called islet amyloid polypeptide (IAPP)) is a pancreatic beta-cell hormone that is co-secreted with insulin in response to nutrient stimuli. The last 35 years of intensive research have shown that amylin exerts important physiological effects on metabolic control. Most importantly, amylin is a physiological control of meal-ending satiation, and it limits the rate of gastric emptying and reduces the secretion of pancreatic glucagon, in particular in postprandial states. The physiological effects of amylin and its analogs are mediated by direct brain activation, with the caudal hindbrain playing the most prominent role. The clarification of the structure of amylin receptors, consisting of the calcitonin core receptor plus receptor-activity modifying proteins, aided in the development of amylin analogs with a broad pharmacological profile. The general interest in amylin physiology and pharmacology was boosted by the finding that amylin is a sensitizer to the catabolic actions of leptin. Today, amylin derived analogs are considered to be among the most promising approaches for the pharmacotherapy against obesity. At least in conjunction with insulin, amylin analogs are also considered important treatment options in diabetic patients, so that new drugs may soon be added to the only currently approved compound pramlintide (Symlin^®^). This review provides a brief summary of the physiology of amylin’s mode of actions and its role in the control of the metabolism, in particular energy intake and glucose metabolism.

## 1. A Brief History of Amylin and Scope of Review

In 1986, two independent research groups identified amylin, also known as diabetes-associated peptide (DAP) or islet amyloid polypeptide (IAPP), as the major component of pancreatic islet amyloid deposits [1,2], which had been a known pathological hallmark of type 2 diabetes (2DM) and feline diabetes since the early 20th century [3,4]. Non-aggregated amylin was then shown to be present in the same secretory granules of the pancreatic beta cells as insulin [5] and was identified as a novel hormone in circulating plasma in 1989 [6]. It was subsequently shown that insulin and amylin were simultaneously secreted from the beta cells in response to glucose and arginine stimuli, and that a selective loss of amylin secretion was observed in rat models of a mild form of type 1 diabetes (1DM) [7]. In viewing 1DM and late-stage 2DM as amylin-deficient conditions, amylin replacement therapy soon became a target for glucose control in patients with diabetes. Because human amylin, by the nature of its physiochemical properties, is prone to aggregation, scientists at Amylin Pharmaceuticals Inc. searched for an analogue that would replicate the physiological actions of amylin, in the absence of the challenges of the native peptide. The front-running analog, pramlintide, was patented by Amylin Pharmaceuticals in 1997 and was heralded as the first anti-diabetic peptide-based drug (Symlin^®^) since the discovery of insulin in 1921. No other amylin based drug has received approval since then, but several newly developed analogs have shown very promising results as anti-obesity and anti-diabetes treatments [8,9,10,11,12,13].

Amylin has mainly been researched in the context of two overarching themes: amylin as a soluble, monomeric hormone, whose physiological role is to control glucose appearance in the blood; and amylin as a constituent of amyloid plaques that eventually lead to the dysfunction and destruction of pancreatic islets that underlie some forms of diabetes. The goal of this review is to summarize the role of amylin in both the etiology and treatment of various forms of metabolic control, including diabetes and glucose control. Following an introduction to the basic properties of amylin and its receptors, we will present some of the foundational literature that demonstrated how the hormone amylin and its analogs can influence eating and blood glucose levels. We will then discuss how amylin and its function is altered in disease states where glucose control is compromised and how both traditional and new amylin-replacement therapies are being tested and applied in clinical populations. Along the way, we will highlight modern approaches to further uncover the structure and function of amylin and its receptors under both physiological and pathological conditions (as well as a target for the treatment of 2DM). We will also provide an overview of a new generation of amylin-based compounds, including DACRAs (dual amylin and calcitonin [CT] receptor agonists) and LAMAs (long-acting amylin agonists), which have more recently revived interest in this second beta cell hormone [8,9,10,14].

## 2. Introduction to Amylin

### 2.1. Amylin as a Pancreatic Beta-Cell Hormone

Amylin is derived from a precursor proamylin [15]. Following eventual cleavage by prohormone convertase 1/3 (PC1/3) and PC2, the 37-amino-acid, the biologically active monomeric form of amylin is formed. The amino acid sequence of amylin displays similarities with CT, α-, β-CT gene-related peptide (CGRP), adrenomedullin (AM) and AM2, which together comprise the CT peptide family [16]. The primary structure of amylin is highly conserved across species [17,18], with the exception of amino acid residues 20–29. While this region does not seem critical for the physiological action of amylin, residues 20–29 constitute the amyloidogenic region that promotes (or prevents) the formation of beta-sheet structures. The presence of multiple proline residues within this region, as is observed in rats and mice, appears to prevent beta-sheet formation [18,19]. Therefore, unlike humans, primates and cats, non-transgenic rats and mice do not develop amylin amyloids and, therefore, lack this pathological hallmark of 2DM (reviewed in: [20,21]). In an effort to understand the cause and effect of amyloid plaques, rat and mouse models have been engineered to overexpress human amylin (hIAPP; e.g., [22,23]) and the prevention or removal of these plaques may become a new anti-diabetic treatment strategy [24].

Expression levels of amylin and insulin are controlled by common promoter elements and the hormones are stored in and co-secreted from the same islet secretory vesicles at a ratio of approximately 1:100 of amylin to insulin [25]. Secretion from the islets is stimulated by glucose, arginine, and fatty acids [26,27,28,29], following a similar molar ratio [30]. Thus, in healthy individuals, amylin levels in the plasma are lowest during a fast (5–10 pM) and peak at around 20 pM after a meal [31]. The influence of various metabolic disease states on basal and stimulated amylin levels are discussed in Section 5.

While amylin was first thought to be synthesized exclusively in the pancreas [32], it was later shown that synthesis also occurs in other parts of the gastrointestinal tract and in spinal ganglia [33,34,35,36,37]. More recently, several independent groups have also identified amylin mRNA and protein in discrete areas of the brain. Amylin was first shown to be dramatically upregulated in the medial preoptic area of lactating rat dams [38] and later found to be highly co-expressed with prodynorphin-expressing neurons in the lateral hypothalamic area [39]. The latter study demonstrated that mRNA levels of amylin varied widely with sex, diet composition, and leptin levels [39]. Still, the functional significance of this centrally produced amylin has not be fully determined but it has been suggested that amylin signaling in the medial preoptic area may be involved in the control of maternal behavior [40].

### 2.2. Amylin Receptors and Pharmacology

Like other peptides of the CT family, amylin acts on class B G-protein-coupled receptors (GPCRs) [41]. In mammals, there are two GPCRs which the peptide family acts on: the CT receptor (CTR) and the CT receptor-like receptor (CLR, also known as CRLR). These receptors can be further modified by receptor activity-modifying proteins (RAMPs) [42,43]. In humans, rats, and mice, there are three known RAMPs which enable different receptors for this peptide family: CLR and RAMP1 form the CGRP receptor and CLR, together with RAMP2 and 3, forms AM1 and AM2 receptors, respectively. CTR acts as a CT receptor but forms amylin receptors when co-expressed with RAMPs 1, 2, and 3, which are named AMY1, 2, and 3 receptors, respectively. Unlike CTR, CLR alone is not a known receptor on any cell surfaces as there is no known ligand to it.

The expression of the necessary AMY components CTR plus RAMPs has mainly been studied in the brain and their presence has been described in multiple brain areas [14,44]. Surprisingly, only very few studies have tested the expression of these components in the same cell, which, according to our understanding, is necessary to form fully active AMY receptors. Such co-expression in the same cell has been shown for the area postrema (AP) in the caudal hindbrain and in parts of the hypothalamus [45,46]. Further, we are not aware of studies characterizing AMY receptors with all their receptor components in the periphery since the first description of RAMPs in 1998 [42]. Hence, whether the peripheral effects of amylin that have been described early after the discovery of amylin (e.g., induction of insulin resistance, inhibition of insulin secretion; summarized in [47,48,49]) are truly mediated by the AMY awaits further investigation.

#### 2.2.1. Receptor Agonism of CT Family Peptide Receptors

Amylin shows low affinity to CTR, but if associated with the RAMPs, CTR forms AMY receptors and responds preferentially to amylin [43,50]. There are species differences in structure and receptor affinity of CT. Salmon CT (sCT) seems to be a potent agonist of both CT and amylin receptors [51] and can therefore be seen as a natural dual receptor (AMY and CTR) agonist. While rat CT is a much weaker agonist of all amylin receptors [50], human CT, on the other hand, shows similar potency to CTR and AMY1 receptors, but a 10-fold lower potency to AMY3 receptor [43]. Due to these findings, the fact that AMY receptors expression requires CTR and the lack of subtype specific receptor antagonists it is currently not possible to test the activation and effect of AMY and CT receptors separately using pharmacological approaches [43].

Furthermore, CT/CGRP family peptide receptors are also, at least partially, responsive to many different peptides of this family. For example, although amylin is a very weak CGRP receptor agonist, AMY1 and AMY3 receptors show high affinity towards CGRP binding [50,52]. In rodents, it was shown that rat α-CGRP displayed a similar affinity to rat AMY1 and AMY3 receptors as rat amylin. Both peptides also showed a partial agonism towards rat CTR, which was 20-fold weaker than rat CT [50]. In contrast to findings in rodents, human α-CGRP showed a lower potency than amylin on a human AMY3 receptor [43]. Rat CT acts 10-fold weaker than rat amylin and rat α-CGRP on AMY1 and AMY3 receptors [43]. Another difference between both species was the comparison between α and β-CGRP: while rat β-CGRP displayed lower potency on rat CT, AMY1, and AMY3 receptors compared to rat α-CGRP [50], human β-CGRP was equipotent to human α-CGRP on human AMY3 and even more potent on human CT and AMY1 receptors [43].

As demonstrated by these findings, it remains challenging to 1. test the response of individual AMY receptors to amylin, 2. to assess amylin’s effect in the absence of other CT family peptides in vivo, and 3. to translate knowledge gathered from animal experiments into clinical applications in humans, in particular when it comes to specificity for certain receptor subtypes. In fact, so far, it can be assumed that some of amylin’s effects observed in vivo may also be partially due to other peptides and receptors [53]; this may be particularly true for brain-mediated effects because CGRP is a wide-spread neurotransmitter in the central nervous system [54]. As will be discussed later, it is interesting to note that the most promising candidates of amylin analogs for the treatment of metabolic disease are, in fact, rather unspecific (dual) CTR and AMY agonists (see Section 6).

#### 2.2.2. Role of Specific Amylin Receptors for Endogenous Amylin Action

Despite the challenges mentioned above, there are approaches to distinguishing the exact function of single CT family peptide receptors. A recent study examined the consequences of the knockout of the RAMP1, RAMP3, and both RAMP1 and RAMP3 genes in mice [55]. It seems that effects requiring RAMP1 are mainly responsible for fat utilization and storage (see also [56]), while RAMP3 is important for controlling glucose homeostasis and food intake. Only mice with a knockout of both RAMP1 and RAMP3 fed on a high-fat diet displayed increased bodyweight, indicating potential redundancy of amylin actions via distinct receptor subtypes. This study provided a first insight into the importance of single amylin receptor components for endogenous amylin action.

An interesting finding provided by multiple studies is that the knockout of RAMP genes, the gene for CTR [55], or administration amylin antagonists lead to metabolic changes, such as increased body weight and food intake [57,58], while mice that are deficient in the amylin gene do not show lasting differences in body weight or weight gain compared to WT mice [59]. The latter also display a higher response to amylin infusion, perhaps indicating a potential up-regulation of amylin receptors and/or amylin signaling pathways. However, it is unclear how the lack of endogenous amylin is compensated for or whether the remaining CT family peptides are involved.

#### 2.2.3. Differences in Internalization and Regulation of Receptor Subtypes

Despite multiple similarities and reported action of CGRP on the AMY1 receptor, a recent study by Gingell and colleagues showed that AMY1 (CTR + RAMP1) and CGRP (CLR + RAMP1) receptors differ strongly in internalization properties [60]. By using fluorescent CGRP and amylin analogs, it was shown that the CGRP receptor is rapidly internalized from the cell surface into the cytoplasm after binding to CGRP analogs, while the AMY1 receptor remains on the cell surface after binding to different agonists. Furthermore, the same study showed that the CGRP receptor was degraded significantly after 4 h of agonist stimulation, while the AMY1 showed no degradation after agonist stimulation [60]. This could be an explanation for the retained amylin sensitivity in hyperamylinemic rats [61]. These findings clearly indicate that there are differences in regulation of and responses to ligand binding. This may be even more important for natural receptor agonists like sCT or synthetic compounds that seem to display irreversible binding to the receptor complex, which may lead to continuing receptor activation [62,63]. Overall, it is clear that further research is needed to assess the exact differences in amylin receptor function, if and how this affects the overall function of the peptides, and how these findings translate to humans.

## 3. Physiological Role of Amylin in Appetite Control

The role of amylin in controlling eating has recently been amply reviewed (e.g., [14,64,65,66]). The most important points will be briefly summarized here. Meal-ending satiation is the major physiological effect induced by amylin in the control of food intake [67]. Amylin dose-dependently reduces the size and length of a meal, with a rapid onset of this effect within minutes [68]. A large number of studies have shown the physiological relevance of this action (e.g., [57,69,70]). Together with amylin’s effect to slow gastric emptying and to reduce pancreatic glucagon secretion, which will be discussed later, the meal size effect of amylin is the most relevant aspect for the regulation of nutrient fluxes [71]; amylin, by controlling the rate of nutrient appearance, complements the action of insulin, which mainly controls the disappearance of nutrients from the primary nutrient pool.

More than three decades of work have resulted in a clear picture about the mechanisms underlying the satiation-inducing effect of amylin and its analogs. When treated acutely with peripheral amylin, rodents demonstrate significant reductions in food intake for approximately two hours [72,73], which primarily results from reduced meal size [68]. Acute or chronic infusion of amylin directly into the brain also significantly reduces food intake and body weight [74], and pre-meal treatment with AC187 leads to an increase in food intake, demonstrating that endogenous amylin contributes to the control of eating [70,75]. Lesion studies have shown that an intact AP, but not vagal or non-vagal afferents, is required for amylin-induced satiation [72,76]. This was confirmed recently in mice whose CTR was depleted in the caudal hindbrain. CTR-depleted mice had a blunted response to amylin injection and, under baseline conditions, the duration of meals was inversely related to the number of neurons expressing CTR [69]. Interestingly, the vast majority of amylin-responsive cells in the AP are also glucose-responsive [77], and it has been proposed that a certain level of blood glucose is permissive for some of amylin’s actions [77,78]. This also seems to apply specifically to amylin’s effect on food intake because our data in rats during hyperinsulinemic euglycemic or hypoglycemic clamp indicated that the acute eating inhibitory effect of amylin was reduced in hypoglycemia (approx. 55 mg/dL [3.0 mmol/L]) [79].

Amylin-induced Fos activation is present in the AP, several downstream brain nuclei, including the nucleus of the solitary (NTS), the lateral parabrachial nucleus (LPBN), and the central nucleus of the amygdala [80]. In addition to blocking the inhibitory effect on food intake, lesions of the AP also reduced the presence of Fos in these downstream nuclei, which supports the idea that the AP is the primary mediator of amylin’s effect on eating [80]; one exception was reported recently because peripheral amylin induced a positive pERK response in the hypothalamus, independent of amylin action in the AP [45,81], but a functional correlate for this effect has not yet been determined.

Collectively, the AP lesioning data and other data indicate that the amylin-induced signal seems to be transmitted via the NTS, the LPBN, and the central nucleus of the amygdala and possibly other higher brain centers [76,82,83,84] (reviewed in [14,44,65]). Several studies showed that the eating inhibitory effect of amylin relies on noradrenaline signaling within the caudal hindbrain. This involves AP neurons that project directly or indirectly to the LPBN [44,83,85]. Direct input from CTR carrying neurons in the NTS [86] with converging projections to the LPBN may also participate in amylin’s effect on eating, but the physiological relevance of the latter projections is not yet clear. Interestingly, nothing is known as yet about the neurotransmitter systems necessary for amylin’s effect in the LPBN and its more rostral projection areas [84]; CGRPergic neurons, which had been suggested previously [87], do not seem to be involved [84].

Despite the pivotal role of the caudal hindbrain in mediating amylin’s effect on eating, amylin and its receptor agonists clearly also affect other brain areas, including the reward system encompassing the ventral tegmental area (VTA) and the nucleus accumbens (NAc). Whether effects in these brain areas are direct or indirect is still under debate (e.g., [88,89,90,91]).

Direct action of amylin and its agonists in various neuronal populations in the hypothalamic arcuate nucleus (ARC) has also been demonstrated [45,92] but such activation has not yet been linked unequivocally to specific amylin actions. We believe that it is likely that they contribute to amylin’s effects to control eating and metabolism, but future studies will need to investigate their exact contribution in this respect, as well as their interaction with caudal hindbrain mechanisms.

One of the best investigated functions of amylin is the role of amylin as a leptin sensitizing hormone. The earliest studies suggesting a functionally relevant interaction between amylin and leptin showed that animals with defective leptin signaling (e.g., ob/ob mice, Zucker fa/fa rats) present a reduced response to the administration of sCT [93] and that central leptin administration increased the effect of peripheral amylin to acutely reduce eating [94]. We confirmed the former results, showing that leptin receptor deficient db/db mice and Zucker diabetic fatty rats are less sensitive to the satiating effect of peripheral amylin compared to wildtype control mice and rats, respectively [95]. Many follow-up studies investigated the type of interaction, mode of interaction, and potential site(s) of interaction between amylin and leptin [96,97,98,99,100,101,102,103]; most of these clearly indicated an important role of the hypothalamus, in particular the ventromedial hypothalamus (VMH), which encompasses the ARC and ventromedial nucleus of the hypothalamus (VMN). This is also consistent with the co-distribution of amylin receptor components and leptin receptors in many brain areas, especially in the hypothalamus [104,105,106,107,108]. Amylin stimulates the production of interleukin-6 (IL-6), selectively in hypothalamic microglia, but not neurons or astrocytes [109]. Because the amylin-induced effects on leptin signaling in the VMH were reduced in IL-6 knockout mice or in rats pretreated with an IL-6 antibody [109], IL-6 may be the mediator increasing the leptin response, as has also been confirmed in dissociated VMN neurons [110]. Furthermore, [111] showed that VMN neurons from diet-induced obese (DIO) rats also express less of the Bardet Biedl Syndrome-6 protein (BBS6) mRNA, which contributes to the trafficking of the leptin receptor to the cell surface [112,113]. Interestingly, IL-6 restored leptin receptor (LepRb) and BBS6 expression in VMN neurons from DIO rats to levels similar to those found in DR rats [110]. Overall, these and other results implied that IL-6 (potentially via BBS6) may link amylin to enhanced VMH leptin signaling [105,109]. Despite the strong evidence for a critical role of the VMH in mediating the leptin sensitizing effect of amylin, some important experiments (e.g., the abolition of the interaction in the absence of specific hypothalamic neurons) have not yet been performed; furthermore, a recent study cast doubt on microglia being the sole source of IL-6 in this respect [114].

## 4. Physiological Role of Amylin in Glucose Control

Since the discovery of insulin in the 1920s, insulin and later glucagon were seen as the major controllers of glucose metabolism. However, in the late 1980s and early 1990s, amylin was recognized as another pancreatic hormone which also influences glucose metabolism [115]. In simple terms, as insulin controls the rate of glucose uptake from the blood, amylin complements insulin by delaying glucose inflow into the circulation [116]. Amylin controls glucose appearance via three primary mechanisms. Amylin decreases gastric emptying, thus delaying the inflow of nutrients into the small intestine, and, by doing so, slows the uptake of meal-derived glucose into the circulation [117,118]. Amylin blocks meal-induced glucagon release, which therefore suppresses glucagon-stimulated hepatic gluconeogenesis [119]. Additionally, as described above, amylin induces meal-ending satiation [72,73,120] and, chronically, also reduces body adiposity and increases energy expenditure [121,122]. In addition to these three main effects, historically being one of the first effects of amylin that was reported, amylin has an inhibitory effect on the secretion of insulin [123,124] and increases the activity of muscle glycogen phosphorylase, thus stimulating muscle glycogenolysis [125]. The physiological or pathophysiological relevance of the latter two effects is, however, still unclear.

### 4.1. Amylin Slows Gastric Emptying

The first study to observe the effect of amylin administration on gastric emptying in humans came from Kolterman and colleagues in 1995 [126]. Diabetes mellitus is often associated with accelerated gastric emptying; subjects with 1DM were treated with pramlintide prior to a meal, along with their usual dose of insulin. When the meal was consumed orally, pramlintide reduced meal-induced glucose excursions; however, pramlintide did not affect glucose appearance after an intravenous glucose load [126]. These data show that while pramlintide decelerated glucose uptake from the meal-derived carbohydrates, it did not affect post-absorptive glucose metabolism.

By tracking the appearance of tritium-labelled glucose in the plasma after glucose gavage, a similar effect of rat amylin and sCT was observed in rats [124,127]. When further probing the mechanism of action in rats, it was shown that both an intact AP [128] and an intact vagus nerve (most likely via parasympathetic efferents) were required for amylin to slow gastric emptying [129]. In obese Zucker rats, a higher dose of amylin was required to observe an effect on gastric emptying [130]. This reduced sensitivity to amylin was postulated to result from the hyperamylinemia—and presumably reduced amylin sensitivity—observed in this model, though it is also possible that the deficiency in leptin receptor which underlies the Zucker rat’s obese phenotype—and hence an attenuated amylin/leptin synergy—could contribute to reduced amylin action [95]. The latter idea receives indirect support from our findings that the acute eating inhibitory effect of amylin is not reduced during prevailing hyperamylinemia [61].

For patient safety, it was important to next determine if this action of amylin could place diabetic patients who are dependent on insulin at a higher risk for insulin-induced hypoglycemia, but it was found that this action of amylin depends on the glycemic state. Treatment with amylin or pramlintide in a normoglycemic state will essentially block gastric emptying during a 20 min test [78]. If amylin was provided along with insulin, which then caused a hypoglycemic state, then gastric emptying was accelerated at a similar rate as when insulin induced hypoglycemia in the absence of amylin. In essence, the presence of hypoglycemia seemed to prevent amylin’s ability to slow gastric emptying, which was proposed as a protective brake to prevent the further deepening of hypoglycemia [78]; similar findings have also been reported in respect to amylin’s eating inhibitory effect [79]. It is possible that this hypoglycemic break on amylin action resides in the brain, as more than 90% of amylin-responsive cells in the area postrema are also responsive to glucose [77].

### 4.2. Amylin Suppresses Meal-Induced Glucagon Secretion

Insulin, amylin, and glucagon work together to control metabolism and blood glucose levels [131]. As glucagon works to mobilize stored energy by acutely stimulating glycogenolysis or gluconeogenesis in the liver, it is released from the pancreatic alpha cells in response to hypoglycemia. Glucagon release is also stimulated by a meal, primarily to buffer and fine-tune the hypoglycemic potential of meal-induced insulin, and it also contributes to meal-ending satiation [132,133]. Diabetes mellitus is often associated with paradoxical postprandial hyperglucagonemia; like insulin, amylin was shown to inhibit glucagon release, which therefore generates a preference for glucose accretion via meal-related glucose over endogenous stores [134]. Notably, amylin does not suppress glucagon secretion in response to hypoglycemia, similar to the fail-safe mechanism discussed above for gastric emptying; in other words, amylin specifically inhibits nutrient- or arginine-stimulated glucagon secretion [135]. Furthermore, this glucagonostatic effect does not occur at the level of the alpha cell, as amylin has no effect on stimulated-glucagon secretion from isolated pancreas or islets [135]. Co-treatment with the amylin receptor antagonist AC187 blocked amylin’s glucagonostatic effect [119,134,136], suggesting that it is likely mediated via amylin receptors, potentially in the AP, however, the precise mechanism or critical site of action has not been determined.

## 5. Role of Amylin in Disease States with Compromised Glucose Control

### 5.1. Type 1 and Type 2 Diabetes

Like insulin, people with 1DM have deficient or absent amylin secretion [31,137]. While people exhibiting glucose intolerance or early-stage 2DM tend to have elevated amylin levels in accordance with hyperinsulinemia [31,138,139], amylin levels at later stages of insulin-deficient 2DM have been described as normal [140] or deficient [139]. Similar effects have been observed at the level of the pancreas. In one study, the majority of beta cell samples collected postmortem from humans with 2DM exhibited no amylin immunoreactivity, while all maintained reactivity for insulin; obviously, this finding must be interpreted with caution because the affinity and sensitivity of antibodies to detect amylin or insulin may differ. Nonetheless, beta cells from subjects without diabetes were positive for both amylin and insulin [141]. Thus, similar to insulin secretion, amylin levels initially increase in response to insulin resistance and glucose intolerance, but progression to 2DM and the eventual failure of beta cell secretion results in a state of amylin-deficiency, which seems to precede, and potentially even contributes to, insulin deficiency [139,142].

These findings are consistent across several rodent models of 1DM and 2DM. [143,144]. For example, Goto-Kakizaki rats, which are nonobese but spontaneously develop 2DM at around 4 months of age, exhibit no increase in plasma amylin following a glucose challenge [145]. Furthermore, streptozotocin, which leads to beta-cell damage and a reduction in beta-cell mass, reduced insulin and amylin expression in rats but, depending on the extent of damage, the amylin to insulin ratio was differentially affected. Specifically, the amylin to insulin ratio was higher when the beta-cell damage was more extensive (i.e., when a higher streptozotocin dose had been used) or when moderate beta-cell damage occurred in combination with insulin resistance. Whether and how changes in the amylin to insulin ratio may affect the pathogenesis or the progression of diabetes, remains unknown, however [146].

Typically associated with 1DM and 2DM are abnormally high glucagon concentrations, in particular in the prandial and postprandial phase. This leads to a paradoxical increase in hepatic gluconeogenesis, which contributes to the hyperglycemic state. The pathophysiological importance of excessive glucagon secretion can, e.g., be demonstrated in insulin-deficient mice with a genetic deletion of the glucagon receptor [147] or rats treated with glucagon receptor antibodies [148]. In both cases, insulin-deficient hyperglycemia could be completely normalized when glucagon signaling was abolished, without any insulin substitution.

Because insulin alone, despite its glucagonostatic effect, often does not lead to optimal glycemic control in particular in the periprandial stage, the observation that amylin also reduces (postprandial) glucagon secretion was the basis of studies investigating the clinical utility of this approach. In fact, it was shown that physiological concentrations of amylin suppressed glucagon release [119,134]. Subsequent studies showed that the effect is of clinical relevance because patients with 1DM or 2DM and exaggerated postprandial secretion of glucagon could successfully be treated by the administration of pramlintide [149,150]. Pramlintide in particular prevented the excessive meal-related rise in glucagon in insulin treated diabetics [151,152]. Importantly, no glucagonostatic effects of pramlintide was seen during hypoglycemia [153], similar to the effect of amylin on eating [79] and on gastric emptying [78]. These are very relevant aspects for drug safety. Hence, the inhibitory effect of amylin on glucagon secretion in rodents could be translated to diabetes treatment and lead to the approval of pramlintide (Symlin^®^) as antidiabetic drug.

### 5.2. Gestational Diabetes Mellitus

Gestational diabetes mellitus (GDM) is characterized by hyperglycemia, resulting from an inability of the maternal pancreas to secrete enough insulin to compensate for insulin resistance that is a normal metabolic adaptation of pregnancy. While glucose-stimulated amylin levels were also shown to increase during pregnancy, a diagnosis of GDM did not seem to have additional influence on increased amylin secretion [154]. Pilot experiments from our group demonstrated that rats in mid-gestation exhibited a normal satiation response to exogenous amylin, suggesting that differently to insulin resistance, amylin resistance does not seem to occur during pregnancy—at least in respect to the eating inhibitory effect [155,156]. While these data would suggest that changes in amylin levels or sensitivity during pregnancy do not contribute to the pathogenesis of GDM, a more recent study showed that pregnant transgenic human-amylin (hIAPP) mice exhibit GDM-like pathologies [157]. Though there is currently no evidence that GDM in pregnant women results from increased deposition of toxic amylin aggregates in the beta cells, this recent study not only presents a new rodent model of GDM, but also introduces the hypothesis that amylin aggregation in women could contribute to GDM during pregnancy or the increased risk for 2DM in the years following pregnancy [157].

### 5.3. Obesity

Most of the recent amylin research concentrated on its eating inhibitory and weight-reducing properties. This has led to several questions with respect to the potential links between amylin and obesity, including (1) whether amylin action could be exploited pharmacologically to combat obesity, and (2) whether a dysfunction of the amylin system may contribute to the development of obesity. There is very good evidence for the former [8,9,10,63,158,159], and this aspect will be covered in the following Section 6.

In respect to the second aspect, amylin, along with leptin and insulin, seems to be an important signal for adiposity [74,75,160], in that it informs the brain about the body’s fat stores. Furthermore, mice lacking the RAMP1 component of the AMY1 receptor are characterized by increased adiposity, and male RAMP1/3 double knockout mice gain more weight when exposed to a high fat diet [55]. Nonetheless, there is currently no clear evidence that a dysfunction in the amylin system (e.g., defects in amylin secretion, changes in receptor distribution or sensitivity) is a primary driver of common forms of obesity. Importantly, high prevailing levels of baseline amylin (e.g., induced by obesity or high fat feeding [143]) do not seem to reduce the efficiency of amylin in reducing eating after acute administration [61]. In other words, we believe that the meal-induced fluctuations of amylin still produce a satiating effect even in obese individuals whose baseline levels are high, and that downregulation of amylin receptors by chronic exposure to high amylin levels may not occur. The latter, however, has not yet been studied formally.

Similar to the correlation of amylin levels with fat mass in rats [143] under weight-stable conditions, basal and glucose-stimulated levels of amylin are elevated in obese men with either normal or impaired glucose tolerance compared to lean controls [139]. The correlation between fat mass and baseline amylin in rats, similar to leptin and insulin, is lost during dynamic periods of weight change, for example, when rats that had been previously forced-fed are returned to ad libitum eating, their fasting amylin levels drop much faster than the reduction in adiposity [161]. However, the functional implications of the finding of this “disconnect” in respect to amylin is unclear.

An important animal model for the study of amylin’s role in obesity in general, and in the amylin-leptin interactions in particular, has been the selectively-bred DIO rat. The intrinsic leptin resistance in DIO rats is already present before the development of overt obesity [162,163]. Leptin resistance is characterized by reduced LepRb expression in the VMH [110,164], reduced leptin receptor binding [104], reduced leptin-induced pSTAT3 [105,163,165,166], and reduced behavioral responses to leptin, e.g., a reduced anorectic effect [162]. Interestingly, lean DIO rats already have reduced amylin binding in the dorsal VMN [105]. It is unclear whether this directly contributes to the reduced leptin sensitivity in DIO rats. However, the depletion of CTR mRNA in the VMH in rats that are diet resistant (DR) reduced their leptin receptor binding and leptin-induced STAT3 phosphorylation. In other words, depletion of the CTR in the VMN rendered a more DIO-like phenotype in the DR rats and led to increased adiposity [105]. Other studies lend support to the important role of CTR signaling on leptin responsive hypothalamic neurons for metabolic control [167,168]. The discovery that leptin receptor deficient db/db mice have less CTR expression in the AP adds to these findings [95], but they cannot easily be translated to humans because primary leptin receptor deficiency is not the cause of common obesity.

Despite the well documented finding that amylin increases leptin sensitivity [94,97,99,100,101,103], amylin administration in rats that were severely obese (approx. 700 g of body weight) produced weight loss (see also [72]), but adding leptin to amylin produced no additional effect [102]. Even when weight loss was induced in these rats by dietary restriction, amylin attenuated the weight regain when the rats were fed ad libitum again, but leptin had no suppressive effect on this weight regain either alone or in combination with amylin [102]. Hence, the synergistic and sensitizing effects of amylin on leptin signaling may reach limits in very obese rodents and possibly extremely obese humans.

It is currently not clear whether the extreme level of obesity or the long duration of high fat feeding, which produces extensive hypothalamic inflammation and gliosis in rodents [169,170,171,172] and humans [170] and subsequently reduced leptin sensitivity and signaling, was responsible for this finding of a missing synergism. It seems possible that leptin resistance due to high leptin levels simply cannot be overcome by amylin administration. The implications for humans are not yet clear, i.e., the effectiveness of amylin–leptin co-administration in obese individuals who are placed on a low fat diet or who were calorically restricted still needs to be investigated.

### 5.4. Bariatric Surgery and Post-Bariatric Surgery Hypoglycemia

Bariatric surgery is currently the most effective treatment of obesity and in many cases results in the rapid improvement of 2DM. Two studies that investigated glucose and mixed-meal stimulated hormone release showed that following Roux-en-Y gastric bypass (RYGB) surgery, but not sleeve gastrectomy or gastric banding, amylin levels were markedly suppressed [173,174]. This is unlike the many other gut hormones, such as GLP-1, PYY, and glucagon, which are typically increased in the weeks and years following RYGB surgery [173,175]. However, some studies in rats indicated that postprandial amylin levels post-RYGB are elevated rather than suppressed [176]. The reasons for these discrepant findings are, as yet, unclear, but study outcome may depend on the time after surgery and on the level of improved insulin sensitivity post-RYGB, hence the strain on beta-cells to secrete insulin (and amylin).

Regardless of the changes of endogenous amylin post-RYGB, amylin was recently discussed in the context of post-bariatric complications. In particular, there is a sub-population of patients who experience post-bariatric surgery hypoglycemia (PBH) in the years following their surgery [177]. PBH has been estimated to occur in between 22 to 75% of patients undergoing bariatric surgery [178,179,180]. It has been suggested that surgical restructuring of the gut after RYGB and the resulting rapid delivery of undigested food into the small intestine causes exaggerated prandial glucose excursions, which leads to a hyperinsulemic response, followed by a hypoglycemic rebound, within one to three hours after the meal [181,182]. While elevated levels of insulin, GLP-1, and glucagon seem even more pronounced in the sub-population most prone to PBH [175], it has not been tested if this same group demonstrates significant changes in the nutrient-induced amylin release compared to the general RYGB population. So, while we do not know if amylin levels are lower in patients with PBH, it was recently hypothesized that amylin and its analogs could be useful in controlling PBH. Based on amylin’s ability to inhibit meal-stimulated glucagon, pre-meal dosing of amylin has the potential to limit glucagon-induced release of endogenous glucose and blunt the amplified glucose excursions observed in patients with PBH. It was hypothesized that suppression of these amplified excursions, including the initial prandial hyperglycemia, could be an important factor in preventing the hypoglycemic rebound that follows a meal. However, a recent pilot study was rather discouraging because pramlintide did not modulate the glycemic excursions or insulin responses in a mix meal test and glycemic excursions remained unaltered [183].

Another unexplored area is whether treatment with amylin or its analogs may be beneficial to curbing weight regain post-bariatric interventions. An important challenge after any weight loss intervention, including bariatric surgery, is the prevention of weight regain [184]. It could therefore be hypothesized that amylin-based pharmacotherapy may be used to prevent weight regain after weight loss. Importantly, it has been shown in rats that had undergone calorie restriction that amylin reduced the weight regain when the animals were again given ad libitum access to food [102]. A translation of these findings could have important clinical implications.

### 5.5. Amylin and Alzheimer’s Disease (AD)

Neurodegenerative diseases, including Alzheimer’s disease (AD) seem to be often associated with metabolic disease, including 2DM [185]. Amylin has recently been suggested as being one potential causal link between these two disease entities, in part because of the biochemical similarities between the amylin-derived islet amyloid and β-amyloid (Aβ) deposition, which is one histological hallmark of AD. Both peptides are prone to aggregation in certain conditions, and the toxic principle of these amyloid aggregates seems to be similar. Furthermore, certain groups of AD patients seem to have amylin-derived amyloid plaques co-localized with Aβ plaques in the central nervous system. Other data, however, suggest that amylin—or at least its non-fibrillar analogs—may reduce Aβ-derived amyloid formation and could in fact be helpful in the treatment of AD. These ambiguities are far from clear, and more research is necessary to better understand the role of amylin in AD and the role of amylin in metabolic changes during ageing in general [186,187].

## 6. Amylin-Based Therapies for the Treatment of Diabetes and Obesity

Diabetes and obesity are both relevant health issues, affecting millions of people all over the world and with an increasing prevalence. Even though lifestyle modification is the first-line intervention for patients suffering, in particular, from 2DM, pharmacological approaches for effective and safe weight loss are needed. Amylin and amylin analogs have been in the focus of interest due to their effects on food intake, body weight, but also on glycemic control. Amylin-based drugs that are currently available or in testing are presented in the following sections.

### 6.1. Pramlintide

Pramlintide (Symlin^®^) was the first approved amylin analogue, and it is still the only approved amylin-based drug so far. Pramlintide only differs in three amino acids from human amylin, which markedly reduces its amyloidogenic properties. Pramlintide has similar potency and biological activity as human amylin and is approved for patients with 1DM and 2DM who use insulin [188,189]. Pramlintide treatment provides a series of benefits such as reduced food intake and body weight, lower glycated hemoglobin level, and it also lowers the insulin dose needed to reach glycemic control in both 1DM and 2DM [189,190,191,192,193]. Additionally, as in the case of amylin, pramlintide slows gastric emptying, inhibits postprandial glucagon rises, and reduces caloric intake, thus delaying glucose appearance in the circulation and controlling glucose homeostasis [118,149,194,195]. While pramlintide was also shown not to compromise the counterregulatory responses to hypoglycemia [196], it can heighten insulin-induced hypoglycemia, and patients combining insulin and amylin therapy are recommended to reduce their normal insulin dose by fifty percent when also taking amylin at mealtimes [197]. Nausea is the most common negative side effect observed in patients taking pramlintide. In addition to contributing to enhanced glucose control in patients suffering from 1DM and 2DM, pramlintide is also considered as a potential drug to treat obesity due to its effect on food intake and body weight [64,198,199], but more modern analogs show greater promise in this respect, not for their main mode of action, but for heightened efficacy and duration of effect because pramlintide requires injections with each major meal, i.e., usually three times daily [192].

### 6.2. Dual Amylin and Insulin-Based Therapies for the Treatment of 1DM and 2DM

Pramlintide is not used as single treatment but is prescribed as an adjunctive treatment in combination with insulin. In fact, the administration of pramlintide plus insulin in a fixed molar ratio improves glycemic control in 1DM [200,201] and also 2DM. Traditionally, insulin and pramlintide had to be administered separately because the coformulation of both peptides was unstable; potentially, this may be a reason why the combination of the drugs is used less frequently than may be indicated. A recent study indicated that a novel approach using a coformulation of supramolecularly stabilized insulin and pramlintide may be possible and that the coformulation was stable over several days; its administration was shown to enhance mealtime glucagon suppression in diabetic pigs [202]. To our knowledge, clinical trials with this new coformulation have not yet been published.

Another indirect approach has been described recently. Insulin-degrading enzymes (IDE) not only metabolize and degrade insulin but also amylin. A recent study showed that the inhibition of IDE modulated the activity of both insulin and amylin in a way that glycemia was markedly improved in preclinical mouse models of metabolic disease [203]. Respective clinical trials have not yet been performed; hence it is unclear if this approach will show therapeutic efficacy.

### 6.3. Long-Acting Amylin Agonists and Dual Amylin and CT Receptor Agonists (DACRAs)

Even though pramlintide shows many benefits in patients with diabetes and obesity, its short half-life and potency limit its efficacy as a medication because it needs to be administered three times daily, i.e., with every main meal [190,192,204]. Thus, more potent and long-lasting amylin analogs have been developed and tested in an effort to provide greater therapeutic benefit.

Davalintide is an amylin analogue with higher potency, efficacy, and a longer half-life [205]. Davalintide shares 49% amino acid homology with amylin and, like amylin and pramlintide, davalintide shows high affinity for amylin receptors, but at the same time it also displays affinity for CT receptors [205]. Preclinical studies in rodents showed that davalintide led to a prolonged receptor activation and also caused a greater reduction of food intake and body weight compared to amylin and pramlintide [205,206]. Additionally, davalintide administration in mice provided glucoregulatory effects such as decreased fasting glucose level [206]. The higher potency is most likely due to the slow dissociation of davalintide from the receptors, similar to sCT [63], as it has a similar circulating half-life as amylin [205,206].

Another approach to enhancing the action of amylin analogs is by modifying their structure. A prolonged half-life can be reached by modifications such as the coupling of a polyethylene glycol (PEG) or by glycosylation. These modified peptides showed prolonged action, though further research is needed to assess their therapeutic utility [207,208].

Recent studies confirmed that the dual agonists, which are able to activate both amylin and CT receptors (DACRAs), similar to sCT, displayed higher potency in the reduction of food intake and body weight compared to amylin and current amylin analogs [209,210,211]. DACRA is a working class of molecules which activates both amylin and CT receptors in a prolonged fashion [209,210,212]. Recent studies showed that DACRAs were superior in terms of typical amylin-induced effects as reduction of body weight, food intake, glucagon secretion, and gastric emptying rate when compared to amylin and other amylin receptor analogs [12,210,211]. Furthermore, DACRA treatment resulted in improvement of glucose homeostasis in obese rodents, shown by reduced fasting blood glucose, lower glucose levels after oral glucose tolerance test, lower glycated hemoglobin, and lower insulin levels [209,210,213]. Altogether, preclinical studies in animals showed that DACRAs display higher potency in eliciting classical amylin-mediated effects compared to amylin and amylin analogs and result in improved glucose homeostasis. Therefore, DACRAs have recently been in the focus of research as a potent drug in obesity and diabetes.

The first known DACRAs was salmon CT (sCT), which acts on both amylin and CT binding sites [51]. Metabolic effects of sCT seem to be, at least partially, due to amylin receptor activation [82,124,214]. Interestingly, rat CT does not exhibit high affinity to amylin receptors [214]. As sCT was at first only considered and used as an amylin analogue, many of its effects have been considered as amylin receptor effects, which has yet to be decisively proven. Current studies are also investigating the effects of various engineered DACRAs, including KBP-042, KBP-088, and KBP-089 [11,13,209,215,216,217]. KBP-088 has been directly compared to davalintide [210] and amylin [12] and was superior in both cases in terms of food intake and body weight reduction and improved glucose control. It is important to note that these experiments were conducted in animals, thus it is still unclear what effect DACRAs will elicit in humans suffering from metabolic disease.

### 6.4. DACRA Mechanism

Even though many studies have examined the beneficial effect of DACRA in metabolism and glucose control, little is known about its exact mechanism. Studies with sCT suggest that the metabolic effect of DACRA and its inhibition of gastric emptying are mediated, like amylin, by neurons in the AP, as sCT administration results in prolonged excitation of these neurons compared to amylin [82,214,218]. In vitro studies found that DACRAs activate both CT and amylin receptors potently but that DACRAs show low affinity to the CGRP receptor [13,209]. However, a link between the receptor activation in vitro and DACRA’s effect in vivo has not been established. Recent studies by Larsen et al. reported that some changes in glucose metabolism, such as an improvement in glucose tolerance and a reduction in fasting blood glucose, could only be achieved by using DACRA or amylin and CT analogs simultaneously, but neither by amylin nor by CT analogs on its own [211]. The authors of this study concluded that the CT receptor is needed for DACRA’s full action on glucose control, while it contributes little to other effects such as body weight and food intake reduction. However, the same study also showed that the co-activation of amylin and CT receptors by using amylin and CT analogs were not enough to modify other parameters of glucose metabolism, such as glycated hemoglobin or plasma insulin level, though DACRA effectively changes these parameters. Thus, further research is needed to explore how exactly the CT receptor, independently of the amylin receptor, contributes to DACRA’s effect in metabolism and glucose control.

### 6.5. Specific Role of the CTR in Metabolism

The results of these DACRA investigations suggest that CT and CTR may play a specific role in metabolism. Even though CT was discovered 60 years ago, its exact physiological significance in controlling metabolism is unclear and the importance of CT and the CTR in metabolism is still incompletely defined. Bartelt and colleagues showed in 2017 that CTR knockout in male mice on a high-fat diet led to negative metabolic changes, such as impaired glucose tolerance and higher serum lipids, suggesting the importance of CTR in metabolic control [219]. The same study, however, also showed that a knockout of Calca, the gene coding for CT and CGRP, resulted in improvements in metabolic parameters such as lower fasting glucose, improved glucose tolerance and decreased white adipocyte tissue weight. These improvements seem mainly due to the absence of CT, as a CGRP knockout alone only merely improved glucose tolerance and lowered the weight of white adipocyte tissue to a much lower content [219]. These findings suggest that CTR signaling contributes to improved metabolic control while CT seemed to be detrimental for metabolic health. The latter has also been confirmed by Nakamura and colleagues who showed in 2018 that mice with global knockout of the CT gene displayed lower body weight, decreased serum lipids and visceral fat, lower fasting glucose, and higher insulin sensitivity [220]. It is unclear why the absence of CTR seems to be deleterious to the metabolism, while the absence of its natural ligand CT may lead to improved metabolic control. In order to establish further knowledge about and the exact mechanism of DACRAs, it is necessary to answer this question.

### 6.6. New Developments of Amylin Analogs

One of the most modern and most potent agonists, which is also a DACRA, is the rather unspecific CTR and AMY agonist AM833 (cagrilintide). Cagrilintide produced sustained and marked weight loss when given alone; when combined with the GLP-1 agonist semaglutide, AM833 produced even stronger responses that exceeded the effect of most other pharmacotherapies [158,221,222,223]. Studies of the exact mechanism of action of this and other long acting amylin analogs are ongoing [8,63,218,221,222,224,225] and at least some of the effects seem to be mediated by the necessary interaction with specific amylin receptor subtypes [218,225]. All results obtained so far allow us to conclude that amylin-based weight loss pharmacotherapy can be considered an interesting strategy for the further development of highly effect weight lowering agents.

## 7. Conclusions

This review provides a brief update of the status of the pancreatic hormone amylin as a key player in the control of energy and glucose metabolism. Amylin’s main physiological actions, the reduction of eating, the inhibition of gastric emptying, and the inhibition of glucagon secretion are mediated by the caudal hindbrain. At least in the case of the eating inhibitory effect, we have learned a lot about the underlying brain mechanisms and interactions with other hormones. Soon after the discovery of these effects, it became clear that amylin analogs may be interesting therapeutic targets for the treatment of metabolic disease, in particular obesity and 2DM. Some of these analogs may now be among the most promising future anti-obesity treatments.

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
