# Peer review of "Mediators of Amylin Action in Metabolic Control"

_jcm, 2022, doi:10.3390/jcm11082207_

Round 1

Reviewer 1 Report

This is a nice, updated and well-written review that summarizes the role of amylin as a hormone that possesses several metabolic functions, particularly regarding the regulation of food intake and glucose homeostasis. I really enjoyed reading the text that was very informative and clear. My only suggestion is the inclusion of a Figure that could summarize the main aspects discussed in the text.

Author Response

Reviewer 1:

This is a nice, updated and well-written review that summarizes the role of amylin as a hormone that possesses several metabolic functions, particularly regarding the regulation of food intake and glucose homeostasis. I really enjoyed reading the text that was very informative and clear.

We thank the reviewer very much for these kind words.

My only suggestion is the inclusion of a Figure that could summarize the main aspects discussed in the text.

Answer: Given the short time given to us to revise the paper, we decided to concentrate on the text and not to include a Figure. We hope that the reviewer will accept this point.

Reviewer 2 Report

Dear Authors

The topic of amylin is exciting. The work is prepared carefully and divided into clear subchapters, making it easy to learn about the properties of amylin on different levels. 

English needs minor proofreading. Minor language errors are noticeable. It is worth considering the frequently used word "analogues", which is not an American word, and maybe replace it with "analogs".

One could also add information about the role of amylin in Alzheimer's disease. There are some new and interesting articles on this topic, for example:

Mietlicki-Baase EG. Amylin in Alzheimer's disease: Pathological peptide or potential treatment? Neuropharmacology. 2018 Jul 1;136(Pt B):287-297. doi: 10.1016/j.neuropharm.2017.12.016. epub 2017 Dec 9. PMID: 29233636; PMCID: PMC5994175

or

Grizzanti J, Corrigan R, Servizi S, Casadesus G. Amylin Signaling in Diabetes and
Alzheimer's Disease: Therapy or Pathology? J Neurol Neuromed (2019) 4(1): 12-16

Best regards

Author Response

Reviewer 2:

The topic of amylin is exciting. The work is prepared carefully and divided into clear subchapters, making it easy to learn about the properties of amylin on different levels. 

We thank the reviewer very much for these kind words.

English needs minor proofreading. Minor language errors are noticeable. It is worth considering the frequently used word "analogues", which is not an American word, and maybe replace it with "analogs".

The text was checked carefully and revised.

One could also add information about the role of amylin in Alzheimer's disease. There are some new and interesting articles on this topic, for example:
Mietlicki-Baase EG. Amylin in Alzheimer's disease: Pathological peptide or potential treatment? Neuropharmacology. 2018 Jul 1;136(Pt B):287-297. doi: 10.1016/j.neuropharm.2017.12.016. epub 2017 Dec 9. PMID: 29233636; PMCID: PMC5994175
or
Grizzanti J, Corrigan R, Servizi S, Casadesus G. Amylin Signaling in Diabetes and
Alzheimer's Disease: Therapy or Pathology? J Neurol Neuromed (2019) 4(1): 12-16

This is an interesting point and we added a short chapter 5.5 on this aspect.